# Fake News Reaching Young People on Social Networks: Distrust Challenging Media Literacy

**Ana Pérez-Escoda \*** , **Luis Miguel Pedrero-Esteban** , **Juana Rubio-Romero** **and Carlos Jiménez-Narros**

Department of Communication, Antonio de Nebrija University, C. de Sta. Cruz de Marcenado 27,
28015 Madrid, Spain; lpedrero@nebrija.es (L.M.P.-E.); jrubio@nebrija.es (J.R.-R.); cjimenez@nebrija.es (C.J.-N.)
\* Correspondence: aperezes@nebrija.es

**Abstract:** Current societies are based on huge flows of information and knowledge circulating on the Internet, created not only by traditional means but by all kinds of users becoming producers, which leads to fake news and misinformation. This situation has been exacerbated by the pandemic to an unprecedented extent through social media, with special concern among young people. This study aims to provide significant data about the youngest generation in Spain (Generation Z) regarding their media and information consumption, their social network use, and their relationship with fake news, all in relation to the feeling of reliability/trust. Focusing on a convenience sample of 408 young Spanish students from Generation Z aged 18 to 22, a descriptive exploratory study is presented. Data collection was performed with an adapted questionnaire. Results show that young Spanish people use networks for information, showing a surprising lack of trust in social networks as the media they consume the most. The content they consume the most since the occurrence of COVID-19 is related to politics, entertainment, humor, and music. On the other hand, distrust of politicians, media, and journalists is evident. The conclusion is that media literacy is still more necessary than ever, but with the added challenge of mistrust: maybe it is time to rethink media literacy.

**Keywords:** misinformation; fake news; social media; media; consumption; Generation Z; young people; media literacy

## 1. Introduction

Since COVID-19 was first detected in December 2019 in Wuhan (China), the entire world has struggled with an unprecedented crisis affecting all orders of human life: social relations, economy, labor market, industry, entertainment, journalism, and education [1,2]. Economies around the world have been affected, and stock markets in all countries have suffered losses. In this situation of precariousness and uncertainty, the need for information is growing disproportionately and is overwhelmed by an exponential growth of different types of disinformation flooding the networks. It is generally accepted that this situation has exacerbated existing problems related to misinformation and fake news, with a new phenomenon developing that the World Health Organization (WHO) has called an "in-fodemic" [3]. Current societies are based on huge flows of information and knowledge circulating on the Internet, created not only by traditional means (media communication) but by all users becoming producers [4,5]. Since social networks emerged at the beginning of the present century, the status of information and knowledge has been transformed, experiencing a remarkable change that has implied a wider online environment enhanced by all users. This new information ecosystem [6,7], providing more freedom in a communi-cational manner, has presented from the beginning a major problem: allowing people to spread misinformation without surveillance has promoted an information disorder that is difficult to manage and control [8,9]. The implications of these changes can be highlighted from two different perspectives: (1) referring to democratic societies in terms of trust not only in politicians and institutions but also in media and journalism [10] and (2) in terms

of digital literacy necessities that have arisen from media literacy, since the high-speed information free-for-all on social media platforms and on the Internet has emerged as the main environment for information to circulate [11]. On one hand, the first issue demands specific actions from policy-makers and media outlets trying to fix the problem, firstly through new regulations and secondly through the practice of trustworthy media discourses engaging citizens in order to tackle distrust and democracy erosion [12]. On the other hand, focusing on audiences/citizens, it seems important, more than ever, to provide them with the suitable digital literacy that enables people to interpret and evaluate received information.

Besides this framework, of an intensive spreading of information on social networks, whether it is trustworthy or not, it is important to note a generational aspect: the youngest generations find their natural habitat in social media. Social networks have emerged as the prevailing setting for socialization, information, and entertainment, including education [13]. Their proliferation among the youngest population emerges as an unprecedented social phenomenon (penetration data), so the problems arising from the growth of misinformation and the lack of adequate digital literacy are further accentuated among this population, which spends most of its time interacting on social networks [14].

This framework justifies the main objective of the presented research: to provide significant data about the youngest generation in Spain (Generation Z) concerning their media and information consumption, their social network use, and their relationship with fake news, all in relation to the feeling of reliability. The research aims to contribute from a media literacy training perspective and a media transformation perspective, both contributing to tackling the challenge of misinformation undermining democracy.

## 2. State of the Art

### 2.1. Fake News, Infodemic, Media, and Social Media

The concept of fake news itself is nothing new; as Burkhardt wrote in 2017 [15] (p. 5), "the ability to have an impact on what people know is an asset that has been prized for many centuries". The particular issue regarding fake news in the 21st century is the large possibility of impact and spread offered by social networks. This phenomenon has been defined using different terms and from different perspectives: fake news, misinformation, information disorder, disinformation, and post-truth. The European Commission [16] defined the word "disinformation" as "false, inaccurate, or misleading information, presented and promoted to obtain revenue or intentionally cause public harm", while other authors [17] point out that "fake news" has been chosen as word of the year in British dictionaries such as Collins and Oxford, which define it as false, often sensationalist, information disseminated under the guise of news. The proliferation of this type of news is a problem that affects all citizens, but particularly young Spaniards, who tend to rely on social networks to keep themselves informed, as indicated by Mendiguren, Pérez-Dasilva, and Meso-Ayerdi [18].

On 31 March 2020, the Director-General of the World Health Organization (WHO), Tedros Adhanom Ghebreyesus, stated "we're not just fighting an epidemic; we're fighting an infodemic" [19], referring to news that spreads more easily and faster than the virus. Although this phenomenon has usually been linked to misinformation, the concept indeed has a wider scope. The WHO has defined "infodemic" as "an excessive amount of information about a problem, which makes it more difficult to identify a solution". The WHO's definition outlines one of the major problems of misinformation: an excessive amount of information, true or false, that is all equally reachable. Social networks have frequently raised the level of noise, and in this sense, some studies point out that in critical situations such as the current one, traditional media offer more trust and credibility [8], although new digital media provide a more rapid response to information queries. The digital media in which we are immersed have allowed any citizen to become a speaker of current affairs, due to the great ease with which users generate and distribute content on different platforms as previously studied by Adoni et al. [20]. The traditional media are

no longer the only channel for obtaining information [21]; social networks have become perfect ally for users to quickly find the useful information they need. The consolidation of the Internet and the incorporation of social networks have even modified the traditional agenda-setting theory where the media were the only ones to select the most important news of each day [22]. In the new digital environment, social media have entered the scene, and Facebook, Twitter, or Instagram determine, in many cases, the rundown of a news item or the page composition of a newspaper.

In this sense, social networks, due to their horizontal, multidirectional, simultaneous, and unfiltered nature, pose a challenge for the traditional media, as there are now more actors capable of producing and disseminating content. For the first time in history, journalists and citizens have the same tools at their disposal [13,14,22].

According to Nielsen et al.'s report [1] from 1 January 2020 to 7 June 2020, the Spanish media published 1,138,364 news items, 90.4% of which were published during the months of March, April, and May. Faced with such an avalanche of information, a confused and anxious society is generated, where citizens were forced to set filters that allowed them to access a volume of content that was more digestible for them. However, these same filters also make them potentially more vulnerable to misinformation [23,24]. Public broadcasters were quick to provide a full schedule of content on the pandemic crisis, which translated into public trust. The International Fact-Checkers Network (IFCN) verified more than 6000 fake news during this period [25], acknowledging that the biggest problem for users was that they did not realize they were consuming or sharing fake news. In this regard, a research study from the Washington Post, the New York Center for Social media and Politics, and the Stanford Cyber Policy Center confirmed that readers had difficulties identifying if news content was true or false [26].

*2.2. Focusing on Generation Z*

In this framework, the previously described media consumption seems a relevant issue, focusing on the youngest, who are considered the population more exposed to digital media [4]. Generation Z, also known by different names (centennials, post-millenials, iGen, Gen Zers) has been the subject of growing interest for some time now, mainly because it is considered the authentically digital generation given that it was the first to be born in a fully developed technological environment. This population niche is composed of young people born between the mid-1990s and the early years of this century, although demographers, sociologists, and academics do not quite agree on the years that this generation exactly comprises.

This generation shares many similarities with the so-called millennials (born between the early 1980s and the mid-1990s). Both generations are part of globalization and the dawn of the digital society. They also share the massive use of devices connected to the Internet, which has affected the way they learn and access knowledge [11,27]. However, there are differences between the two generations derived from the socio-economic context and technological advances in which they were born and socialized [28]; changes that, according to published studies, are affecting the perception of the environment, the prevalence of certain values and their relationship with work. Regarding the differences between millennials and centennials derived from the technological context, research points to the explosion of the Internet, which occurred in 1994, as a milestone separating the two generations, as it transformed the practices of digital interaction. There was an exponential increase in the number of information sources, and a more flexible, shared, and mobile organization and transmission of information. Millennials saw the birth of social networks and some of them came of age during this period, so they are considered the first digital migrants [14]. Generation Z, on the other hand, have never known a world without social networks and in which mobile connectivity is the order of the day (recall that the iPhone was launched in 2007). They are therefore young people "shaped" by these new communication technologies and with the capacity to orient their use towards innovation and the design of their professional lives.

Therefore, the Generation Z is the first generation to have been radically affected by digitalization [29], to the point that, according to Isaac Lee, president of Univision News, it has affected them more than cultural, identity, race, or language aspects, which, on the other hand, makes them the most homogeneous generation of the modern era [30]. They lack a pre-smartphone memory, are on social media more constantly [31], and have had Internet 2.0 technology embedded into their lives. Deliberately false information (fake news) is in itself a source of permanent concern as it is rapidly and extensively disseminated due to the strategy of provoking responses of indignation, fear, and surprise. However, it is of particular concern in the case of young people; on the one hand, because they are the most vulnerable and most exposed to social networks, where this type of information circulates unchecked; on the other, because media literacy is part of the educational curriculum for young people. Many publications have dealt with this issue [18,32,33].

*2.3. UNESCO Media Literacy Response*

The United Nations Educational, Scientific, and Cultural Organization, generally known as UNESCO, has been a pioneer in expanding and developing media literacy and media education as a key issue in current societies. From the Gründwald Declaration (1982), where the main framework was established, and then the Alexandria Declaration (2005), which implied a systematization and more precise definition, through the Conference in Vienna (1999), which considered the digital advances and the new communicative era and, to the UNESCO Paris Agenda (2007), the UNESCO has been a pathfinder to media literacy [34,35]. In 2008, the organization presented the ICT Competency Framework for Teachers focusing on ICT in Education as the result of the "mainstream rollout of computers in schools", introducing Technology Literacy as an essential stage of teacher development [35]. After that, in 2011, UNESCO went further with the AMI Curriculum, combining media and information literacy as prerequisites in the Networked Society for all citizens, but focusing on teachers as leaders in media education. The framework established in the AMI Curriculum introduced nine core indicators to be developed from five key elements:

- Understand the role and functions of media and information in democratic societies.
- Understand and access media contents and their uses, in terms of consumption.
- Critically evaluate media content in the light of media reliability.
- Engage with media for self-expression and democratic participation.
- Review skills (including ICTs) needed to produce user-generated content

From the perspective of media and fake news, UNESCO provided a holistic view of the different developments of misinformation with their International Programme for the Development of Communication (IPDC) "encouraging optimum performance and self-regulation by journalists, as an alternative to the risks of having state intervention to deal with perceived problems in the freedom of expression realm" [36] (p. 11). Earlier in 2007, UNESCO published the "Model Curriculum on Journalism Education" and disseminated it worldwide in nine languages. This general concern continued in the publications from 2013, "Model Curriculum for Journalism Education: A Compendium of new Syllabus" and in 2015, "Teaching Journalism for Sustainable Development: New Syllabi". Since 2015, the UNESCO asked media outlets and journalism to be aware in adapting discourses to the new era, being able to [36] (p. 108):

1. Understand how social media have affected the role and profile of journalism.
2. Understand how social media have changed the process of news production and dissemination and the relationship between journalism and audiences.
3. Explore new business and entrepreneurial models for media industries.
4. Discuss ethical challenges and considerations within this new media ecosystem.

Bringing together this state of affairs, we find the context for our study: the "infodemic phenomenon" and the intensive use of social networks by young people. This study is focused on providing significant data on media consumption, social network use, and fake

news relationships associated with media reliability of the Generation Z. In this regard, the following research objectives (RO) were addressed according to the media literacy key elements established by the UNESCO:

- RO1. Determine young Spaniard´s media and information access and consumption
- RO2. Describe media and social media habits in order to discover good or bad practices.
- RO3. Analyze the level of reliability awarded to media by this population.
- RO4. Outline the Spanish young´s relationship with fake news in terms of reception, distinction, and perception.

## 3. Materials and Methods

### 3.1. Study Design: Variables of Study and Instrument

To answer the research objectives, a quantitative, descriptive, and exploratory methodology was chosen. Data gathering was arranged with the design of an instrument adapted from the report *Media Use in the European Union* [37] and *Digital News Report.es 2020* [38], both focused on media and social media consumption and fake news. The questionnaire was considered as the most appropriate ad hoc design-adapted tool to collect the necessary study variables [39] using the Google Forms tool for this purpose. The definition of the study variables was based on three research constructs related to research questions and according to the previous study by Couldry, Livingstone, and Markha [40]: (1) media consumption, (2) consumption of social networks, and (3) disinformation and fake news. The final questionnaire was the result of a twofold process: (1) first, the team designed an adapted questionnaire in which four different blocks were established: (a) sociodemographic variables, (b) variables related to media consumption, (c) variables related to social network consumption, and (d) variables related to disinformation and fake news (see Table 1); and (2) second, it was sent to a panel of experts for validation (n = 5) in an attempt to seek internal coherence and clarity for the studied constructs. After this phase, in which seven items were eliminated and eight were rewritten, the questionnaire comprised 81 items organized into the abovementioned constructs. The distribution tool for the questionnaire was Google Forms, and consent was collected from each individual before the survey was freely taken.

**Table 1.** Study constructs, variables, and number of items. Own elaboration.

| Study Constructs and Research Objectives (RO) | Variables of Study | Number of Items |
|---|---|---|
| C1. Media and information consumption (RO1 and RO3) | $V_1$. Source of information | 8 |
| | $V_2$. Reliability on sources | 8 |
| | $V_3$. Media type | 5 |
| | $V_4$. Since COVID-19 information consumed | 8 |
| | $V_5$. Average time | 1 |
| C2. Social media consumption (RO2) | $V_6$. Social media used | 8 |
| | $V_7$. Level of engagement | 1 |
| | $V_8$. Favorite social media | 7 |
| | $V_9$. Since COVID-19 information received | 8 |
| C3. Misinformation and fake news (RO4) | $V_{10}$. Fake news reception and distinction | 2 |
| | $V_{11}$. Content more related to fake news | 8 |
| | $V_{12}$. Media spreading more fake news | 8 |
| | $V_{13}$. Source reliability | 6 |

Regarding the type of variables used, it is important to note that all of them were qualitative and categorical, divided into ordinal and nominal ones. The ordinal ones were designed with a Likert scale, with a range of responses from 1 to 5, where 1 means "none or never" and 5 means "always, all, absolutely, or constantly". The statistical analysis was descriptive, based on frequencies and percentages, and conducted with the SPSS package version 24. The internal consistency of the test had a high/good reliability with a Cronbach's Alpha of 0.911, 0.831, and 0.802, respectively, for each construct of the study.

According to Vilches [39], when the Alpha coefficient is >0.90, we can say that the reliability of the instrument is excellent, and if it is >0.80, we can say that it is good.

*3.2. Sample*

The sample, conceived as the set of elements of the population that are asked to participate in the investigation [39], corresponded to undergraduate students from different parts of Spain as shown in Figure 1. The study did not intend to be representative; thus, the snowball sampling technique was applied, achieving a total sample of 408 students aged between 18 and 22 years old (M = 20.94; SD = 3.28). The sample composition was as follows: 30.9% of the sample (N = 126) was male, and 69.1% (N = 282) was female. The distribution was as follows in Figure 1, with most of the sample being from Madrid (N = 145; 35.3%), Valencian Community (N = 60; 14.7%), and Cantabria (N = 56 13.7%).

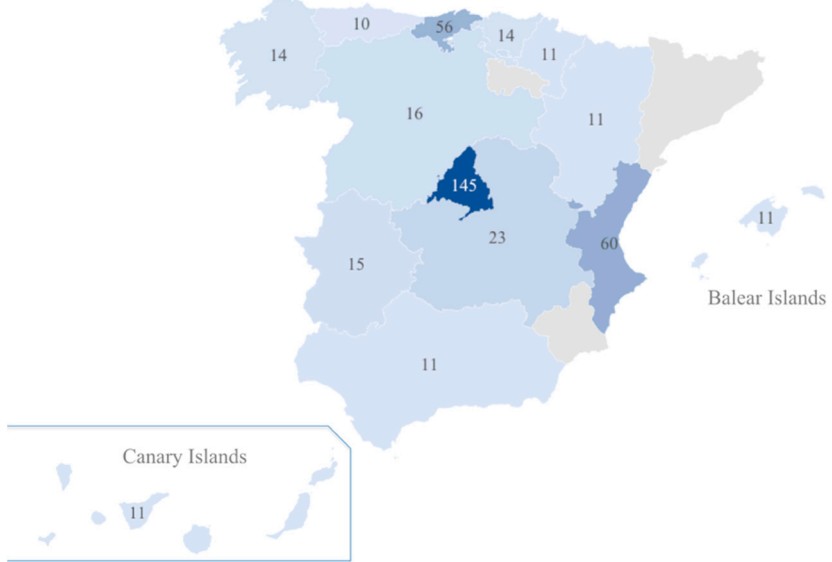

**Figure 1.** Sample frequency distribution. Own elaboration.

The non-probabilistic snowball method used for the sample selection was considered the most suitable for only reaching individuals from Generation Z [41]. The current pandemic situation due to COVID-19, in which the researchers had no possibility of mobility and the students attended schools from home, made this technique the most suitable one. The results were taken from November 2020 to February 2021. It is important to note that the nature of the study, descriptive and merely observational, was appropriate as well, taking into account that no sampling error could be determined and no inferences could be performed from this sampling. However, in order to guarantee the quality of the descriptive research design, the appropriate steps were taken as explained in the previous section.

## 4. Results

The results will now be examined to answer research objectives, corresponding to the three different constructs of study. It is important to note that due to the number of data gathered within the 81 items, only some of the results obtained were analyzed. To present the descriptive analysis results, the data distribution, means, standard derivation, frequencies, boxplots, and crosstabs are used to show the results of the descriptive-explorative study.

*4.1. Media and Information Consumption*

The results found for the first construct of study—media and information consumption—corresponding to four different variables and 29 items in our study, are presented partially

with $V_1$ and $V_2$. The variables were qualitative, categorical, and ordinal with a Likert scale, as can be observed in the following tables. Table 2 shows the results for $V_1$: "I usually get informed with . . . ", where eight different media could be chosen: radio receiver, online radio, press, digital press, TV set, online TV, webpages, and social networks.

**Table 2.** Basic statistics for media normally consume to be informed by Generation Z. Own elaboration.

| I Usually Get Informed with . . . | Mean | DT | Never | A Little | Occasionally | Frequently | Always | N |
|---|---|---|---|---|---|---|---|---|
| Radio | 1.71 | 0.877 | 50.5 | 34.3 | 10.3 | 3.9 | 1 | 408 |
| Online radio | 1.58 | 0.855 | 61.5 | 23.8 | 10.3 | 4.2 | 0.2 | 408 |
| Press | 1.79 | 0.896 | 46.8 | 33.3 | 14.7 | 4.7 | 0.5 | 408 |
| Digital press | 3.42 | 1.083 | 4.7 | 14.5 | 32.6 | 30.4 | 17.9 | 408 |
| TV | 3.5 | 1.145 | 5.6 | 14.5 | 25.7 | 32.6 | 21.6 | 408 |
| Online TV | 2.2 | 1.23 | 39.2 | 23.3 | 21.6 | 9.8 | 6.1 | 408 |
| Web pages | 3.62 | 1.04 | 3.9 | 9.6 | 27.7 | 38 | 20.8 | 408 |
| Social networks | 4.25 | 0.961 | 1.7 | 4.2 | 13.7 | 27.7 | 52.7 | 408 |

The results for the first variable showed us the first insights about media consumption of the Generation Z, with larger differences found for the radio, with a percentage of 50.5 (M = 1.71; DT = 0.877), and the students declaring never getting information from this medium. The same was found for online radio, with a percentage of 61.5 (M = 1.58; DT = 0.855) of the participants; the press, with a percentage of 46.8; and online TV, with 39.2% (M = 2.2; DT = 1.23) of the sample never getting information from it. The opposite was found for the consumption of digital press, reaching 48.3% (M = 3.42; DT = 1.083), in which respondents stated that they frequently or always obtained information from this media; the same was found for webpages, with a percentage of 58.8 (M = 3.62; DT = 1.04) and social media with the highest percentage, 80.4 (M = 4.25; DT = 0.961). It is worth noting that the TV was the only traditional medium that was still utilized by young audiences. In this regard, we found that 54.2% (M = 3.5; DT = 1.145) of our sample declared that they frequently or always used it to become informed.

The second variable analyzed in this construct of study was related to the reliability of the media used to become informed. As shown in Table 3, the results were opposite from the results for the first variable.

**Table 3.** Basic statistics for reliability perception in media consumed by Generation Z. Own elaboration.

| Perceived Reliability of . . . | Mean | DT | Never | A Little | Occasionally | Frequently | Always | N |
|---|---|---|---|---|---|---|---|---|
| Radio | 3.46 | 0.823 | 1 | 12.3 | 32.4 | 48.3 | 6.1 | 408 |
| Online radio | 3.26 | 0.798 | 1.5 | 15.2 | 41.2 | 39.7 | 2.5 | 408 |
| Press | 3.59 | 0.879 | 1 | 11 | 28.4 | 47.1 | 12.5 | 408 |
| Digital press | 3.35 | 0.868 | 1.2 | 15 | 39.2 | 37 | 7.6 | 408 |
| TV | 3.28 | 0.976 | 3.2 | 19.9 | 31.1 | 37.7 | 8.1 | 408 |
| Online TV | 3.04 | 0.942 | 4.9 | 23.5 | 38.5 | 28.9 | 4.2 | 408 |
| Web pages | 2.66 | 0.831 | 4.7 | 41.4 | 38.7 | 13.5 | 1.7 | 408 |
| Social networks | 2.41 | 0.862 | 11.8 | 47.3 | 31.1 | 8.1 | 1.7 | 408 |

The data obtained for this variable showed that the media that was more appealing to the Generation Z were indeed those which they considered to be less reliable. As shown in Table 2, more than a half of the sample considered the radio (54.4%; M = 3.46; DT = 0.823) and press (59.6%; M = 3.59; DT = 0.879) to be frequently or always reliable media, followed by the TV (45.8%; M = 3.28; DT = 0.976), digital press (44.6%; M = 3.35; DT = 0.868), online radio (42.2%; M = 3.26; DT = 0.798), and online TV (33.1%; M = 3.04; DT = 0.942). Lower rates of reliability were found in the most consumed media by the Generation Z: webpages and social networks. Only 15.2% (M = 2.66; DT = 0.831) and 9.8% (M = 2.41; DT = 0.862), respectively, perceived these media as reliable, with these results being certainly surprising, taking into account these were the most used for being informed. If we address these two

variables together in a graphical distribution, the results are very interesting as shown in Figure 2.

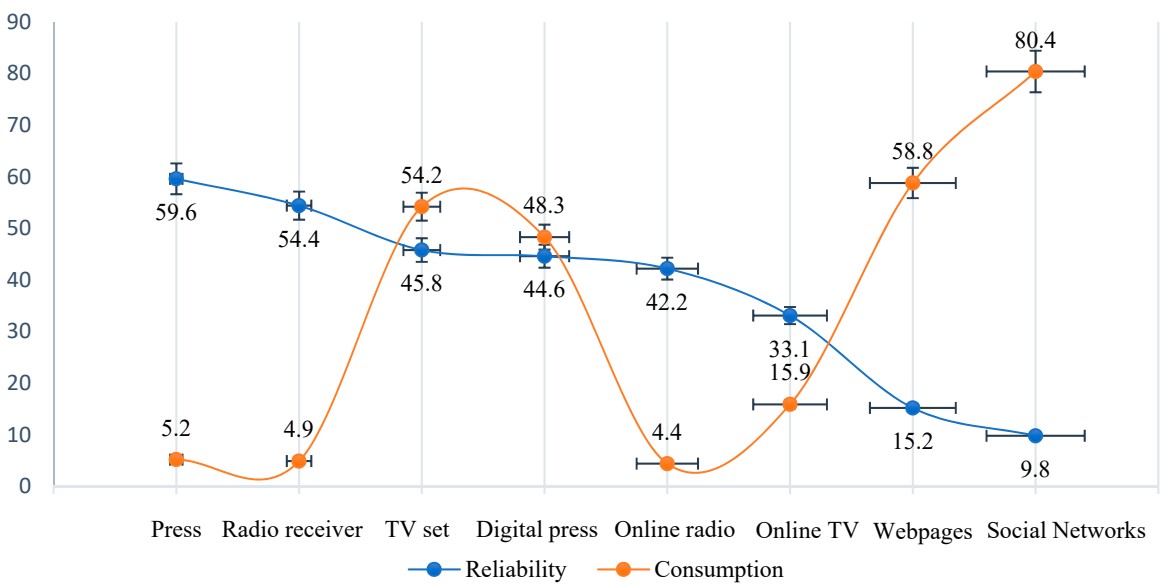

**Figure 2.** Graphical distribution for media consumption and media reliability for Generation Z. Own elaboration.

It can be observed that the less consumed media—the press, radio, and online radio—were perceived as the most reliable, and on the contrary, the most consumed media (webpages and social networks) seemed to be perceived as the least reliable media. To complete an analysis of the media consumption results, it makes sense to ask our sample which kind of information they preferred to become informed about, as observed in Figure 3.

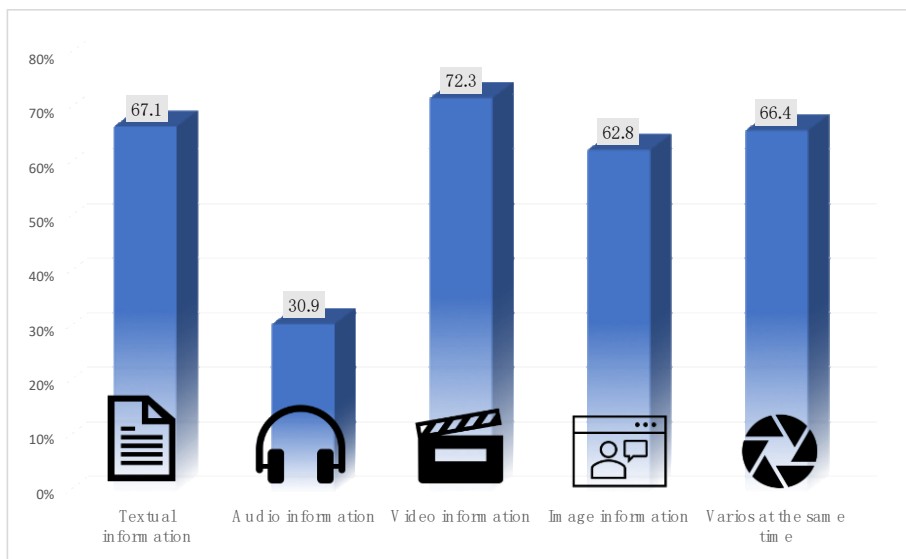

**Figure 3.** Graphical percentages for preferences in data types for information consumption. Own elaboration.

The last variable in the first construct was related to what kind of information was more consumed since the COVID-19 pandemic began. The results are as shown in Table 4, as follows.

**Table 4.** Basic statistics for (V$_4$): what kind of information have consumed the most since COVID-19 occurred? Own elaboration.

|  | Mean | DT | Never | A Little | Occasionally | Frequently | Always | N |
|---|---|---|---|---|---|---|---|---|
| Healthcare | 3.43 | 1.162 | 4.4 | 19.4 | 27.3 | 26.5 | 22.4 | 408 |
| Alternative medicine and self-help | 2.45 | 1.163 | 21.6 | 38.6 | 19.7 | 13.5 | 6.6 | 408 |
| Politics | 4.09 | 1.016 | 2.2 | 5.4 | 18 | 30.3 | 44.1 | 408 |
| Entertainment and Culture | 3.15 | 1.224 | 7.9 | 27 | 24.8 | 22.6 | 17.7 | 408 |
| Food and care | 2.86 | 1.134 | 12.3 | 29 | 26.3 | 25.8 | 6.6 | 408 |
| Sports | 2.88 | 1.221 | 15 | 25.3 | 27.3 | 21.6 | 10.8 | 408 |
| Sexuality and privacy | 2.2 | 1.065 | 28.7 | 38.8 | 20.4 | 8.1 | 3.9 | 408 |
| Humor | 3.83 | 1.183 | 4.4 | 11.8 | 17.9 | 28.3 | 37.6 | 408 |

We found that there were two specific topics that were most consumed: politics were always consumed by 44.1% (M = 4.09; DT = 1.016), which totals 74.4% if we consider the responses "frequently" and "always"; the other topic with the highest results was humor, always consumed by 37.6% (M = 3.83; DT = 1.183) of the sample and frequently and always consumed by 65.9% of the sample.

### 4.2. Social Media Consumption

We now present the results for the second construct of study—social media consumption—corresponding to four different variables and 17 items in our study. Regarding the first variable—average time spent in social media per day—which represents one of the key aspects when measuring media literacy (access and use), the results are shown in Figure 4.

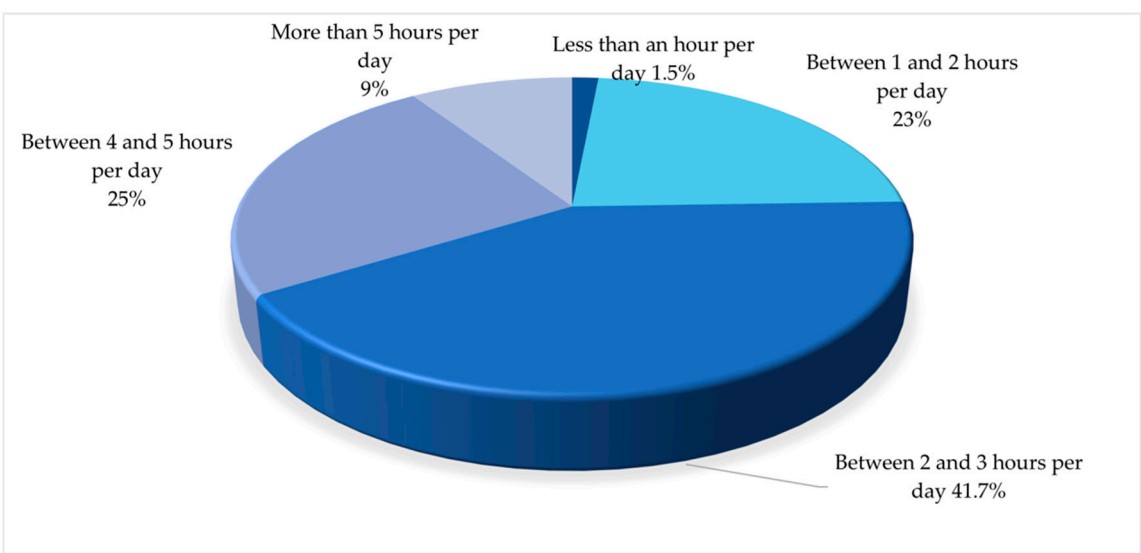

**Figure 4.** Graphical representation average time spend in social networks per day. Own elaboration.

As can be observed, 41.8% of the sample spent between 2 and 3 h per day in social networks, a quarter of the individuals (25%) declared surfing digital media between 4 to 5 h per day and 9% more than 5 h per day. The smallest average was for "Less than an hour per day": only 1.5% declared this range, which indicated that virtually all the individuals in the sample spent time in the social networks. For the next variable of study (V$_6$), we analyzed basic statistics and correlations between the average time spent in social networks and specific social networks, as shown in Table 5.

**Table 5.** Basic statistics for variable (V$_6$) "Define your level of usage for social networks" and correlation with average time of social networks use. Own elaboration.

| State the Level of Use for … | Mean | DT | *p* | Rho | Never | A Little | Occasionally | Frequently | Always | N |
|---|---|---|---|---|---|---|---|---|---|---|
| Facebook | 1.5 | 0.914 | 0.240 | 0.057 | 69.6 | 18.4 | 6.4 | 3.7 | 2 | 408 |
| Tik Tok | 2.26 | 1.418 | 0.000 * | 0.222 | 45.3 | 18.4 | 12 | 13.7 | 10.5 | 408 |
| Twitter | 2.96 | 1.5 | 0.000 * | 0.242 | 23.3 | 22.1 | 13 | 19.1 | 22.5 | 408 |
| Twitch | 1.47 | 0.905 | 0.459 | 0.037 | 72.1 | 16.4 | 6.4 | 2.9 | 2.2 | 408 |
| YouTube | 3.26 | 1.073 | 0.249 | 0.057 | 2.2 | 25.7 | 31.1 | 25.5 | 15.4 | 408 |
| Instagram | 4.12 | 1.02 | 0.000 * | 0.422 | 2.2 | 6.1 | 15 | 30.9 | 45.8 | 408 |
| WhatsApp | 4.32 | 0.943 | 0.003 * | 0.349 | 0.2 | 6.6 | 12.3 | 23 | 57.8 | 408 |

* $p < 0.05$.

From Table 5, it is worth noting that Instagram and WhatsApp were the most used among our sample, with 45.8% and 57.8%, respectively, declaring they used it "always". Regarding correlations between social networks that were most used and average time spent, significant correlations ($p < 0.05$) were found for Tik Tok, Twitter, and Instagram. However, it is important to point out that this correlation is weak in all cases, except for Instagram, showing moderate results ($p = 0.000$; R = 0.422).

According to Voorveld, Guda, Muntinga, and Bronner [42], one of the most interesting issues in the study of social networks consumption is related to engagement as a psychological state of user's motivation, determining different roles adopted when using social networks. As shown in Figure 5 the level of engagement in our sample was different depending on the digital platform. To address this issue, five different levels of engagement were established as defined by Barger and Labrecque [43].

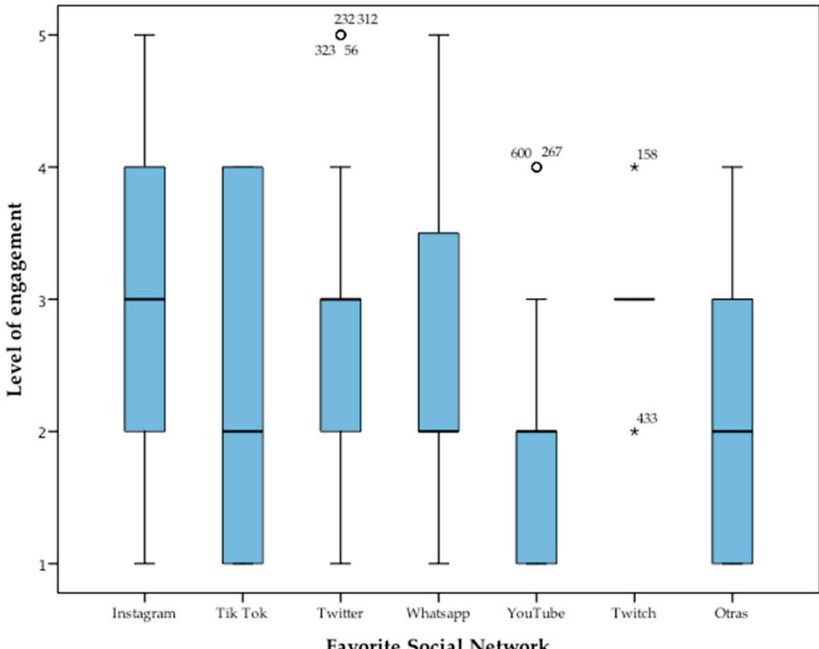

**Figure 5.** Boxplot for distribution of favorite social network and level of engagement in social networks. Own elaboration. Small circles and star values are outliers. Small circles, "out values", and star ones, "far out values".

The levels of engagement were established as follows: (1) I only consume without any participation; (2) I consume and participate by sharing content that I find interesting or that I have created myself; (3) I consume and participate by sharing and commenting; (4) I consume, participate by sharing, comment and seek to mention others; and (5) I consume, participate by sharing, commenting, and seeking controversy/dispute.

The last variable in this construct of the study referred to social media content consumption since COVID-19 began. The results shown in Table 6 for this variable allowed us an in-depth understanding of the variables analyzed.

**Table 6.** Basic statistics for variable (V$_8$): Social media content consumption since COVID-19 occurred. Own elaboration.

| Type of Content | Mean | DT | Never | A Little | Occasionally | Frequently | Always | N |
|---|---|---|---|---|---|---|---|---|
| Entertainment and culture | 4.16 | 0.918 | 0.2 | 5.6 | 16.9 | 32.4 | 44.9 | 408 |
| Fashion and beauty | 2.91 | 1.407 | 21.1 | 22.3 | 18.6 | 20.1 | 17.9 | 408 |
| Information and current issues | 3.38 | 1.045 | 2.7 | 19.1 | 31.1 | 31.6 | 15.4 | 408 |
| Humor and memes | 3.7 | 1.157 | 4.2 | 13.2 | 21.3 | 30.6 | 30.6 | 408 |
| Food and care | 2.36 | 1.183 | 27.5 | 34.3 | 19.6 | 12.5 | 6.1 | 408 |
| Sports | 2.58 | 1.322 | 25.6 | 30.2 | 15.5 | 18.4 | 10.3 | 408 |
| Trending topics | 2.55 | 1.177 | 20.1 | 33.7 | 24.3 | 14.5 | 7.4 | 408 |
| Music | 3.69 | 1.179 | 4.2 | 14 | 22.4 | 27.3 | 32.2 | 408 |
| Cars and motor | 1.52 | 0.99 | 72 | 13.8 | 6.9 | 4.9 | 2.5 | 408 |
| Video games and gamers | 1.97 | 1.291 | 54.3 | 18.4 | 8.8 | 12.5 | 5.9 | 408 |
| Politics | 2.63 | 1.24 | 21.6 | 28 | 25.1 | 16.2 | 9.1 | 408 |
| Challenges | 1.81 | 0.98 | 47.7 | 32.4 | 13 | 4.7 | 2.2 | 408 |
| Healthcare | 2.53 | 1.199 | 23.3 | 29.2 | 26.3 | 13.8 | 7.4 | 408 |
| Technology | 2.38 | 1.185 | 28.5 | 29.5 | 22.6 | 14.3 | 5.2 | 408 |

It is worth noting from these results that the most timely issues in social media consumption for our sample (more than a quarter declared consuming it "frequently" or "always") were "Entertainment and culture", with 77.3%; "Humor and memes", 61.2%; "Music", 59.5%; "Information and current issues", 46%; "Fashion and beauty", 38%; "Sports", 28.7%; and "Politics", 25.3%.

### 4.3. Misinformation and Fake News

The results in the third construct of study—misinformation and fake news—correspond with four different variables and 29 items in our study. The first results addressed were related to fake news distinction and reception, as shown in Figure 6.

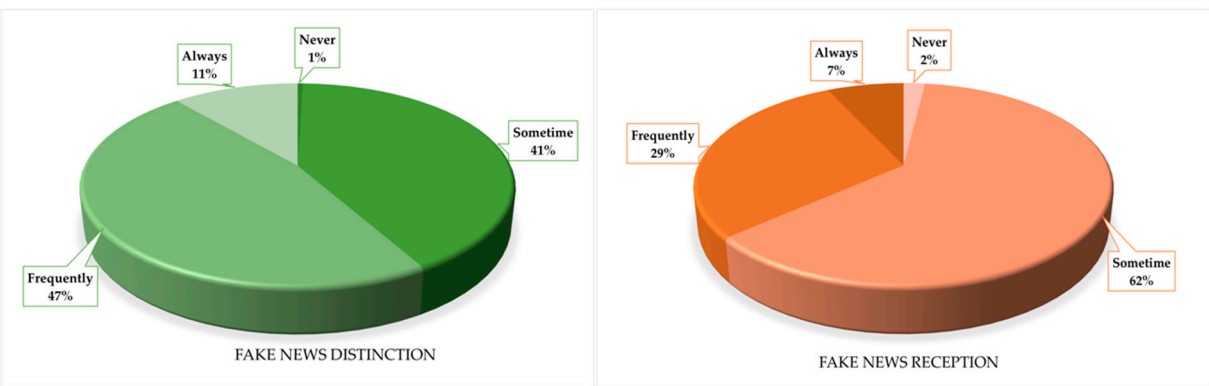

**Figure 6.** Fake news distinction and reception. Own elaboration.

Subsequently, these variables allowed us to analyze, on the one hand, which content was perceived to be more related with fake news (V$_{10}$), and, on the other hand, which media were spreading more fake news (V$_{11}$). Topics for V$_{10}$ were taken from responses obtained in V$_8$: social media content consumption since COVID-19 began. Considering responses from "frequently" and "always", a percentage of 38.9 showed that "Academic content" spread fake news to some extent, and similar results were obtained for contents such as "Sports" with 38.8%, "Beauty and fashion" with 24.4%, "Entertainment" with 33.2%, or "Videogames and gamers" with 27.3%. The results for "Politic content", with

a percentage of 98, and "Humor and gossip" with 83.5%, obtained the highest averages. Regarding variable ($V_{11}$) the results were as follows in Table 7.

**Table 7.** Basic statistics for variable ($V_{11}$): Which contents do you perceive to be more related with fake news? Own elaboration.

|  | Mean | DT | Never | A Little | Occasionally | Frequently | Always | N |
|---|---|---|---|---|---|---|---|---|
| Academic | 2.34 | 0.867 | 11.1 | 57 | 21.1 | 8.6 | 2.2 | 408 |
| Politic | 4.18 | 0.823 | 0 | 2 | 20.6 | 35.1 | 42.3 | 408 |
| Humor and gossip | 3.69 | 1.09 | 2.9 | 13.5 | 21.1 | 36.1 | 26.3 | 408 |
| Health and diet | 3 | 1.109 | 7.6 | 27.8 | 31.9 | 22.1 | 10.6 | 408 |
| Fashion | 2.13 | 0.858 | 20.1 | 55.5 | 17 | 5.7 | 1.7 | 408 |
| Sports | 2.44 | 0.953 | 12 | 49.1 | 24.8 | 10.6 | 3.4 | 408 |
| Entertainment | 2.27 | 0.933 | 19.4 | 47.4 | 21.9 | 9.8 | 1.5 | 408 |
| Videogames and tech | 2.13 | 0.837 | 21.4 | 51.4 | 20.9 | 5.7 | 0.7 | 408 |

As shown in Table 7, results coincided with the perception of distrust and reliability studied in the first construct of the study. It is important to note that the results for Facebook and Tik Tok were not significant as they were already analyzed and our sample did not consume them. At this point, feelings associated with fake news were explored in the next item, as shown in Figure 7.

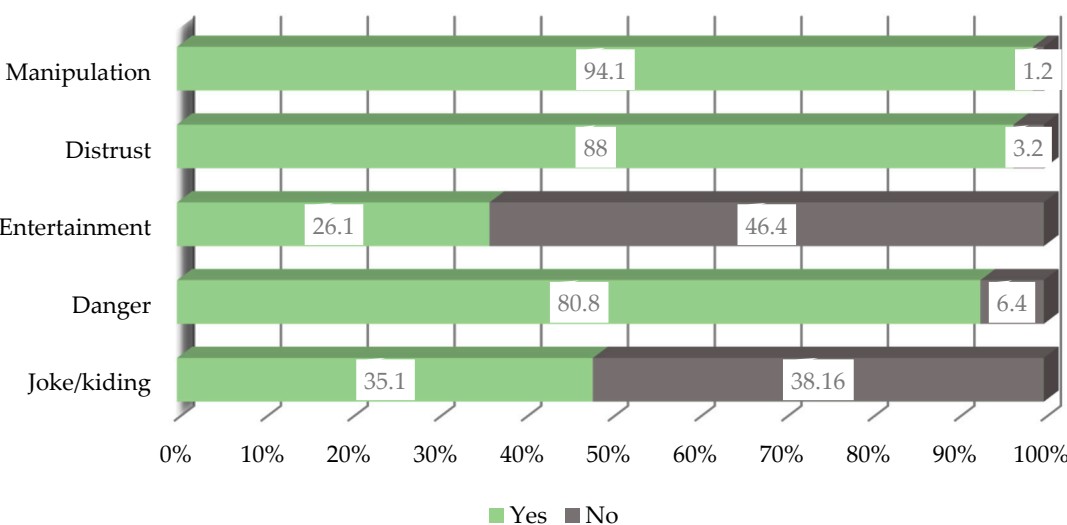

**Figure 7.** What feelings do you associate fake news with? Own elaboration.

Considering the individuals from our sample, we found the following feelings related to fake news: a percentage of 94.1 declared that they associate manipulation with misinformation, 88% stated associating it with distrust, and 80.8% said it was associated with danger. The next step was focused on finding out if they were using fact-checking as a specific tool to fight disinformation and fake news. Although this concept is usually related to journalism, it has experienced an increasing relevance since social media emerged as sources to become informed among the population, and more so since COVID-19 began, as described by Magallón-Rosa [44]. In this regard, we found that 61.2% of the sample declared they did not know what a fact-checker was, and only 38.8% declared that they knew what they were. From this percentage, only 20.9% stated they had used it at least once. In this regard, Newtral, Efe Verifica, and Maldita.es were the most used fact-checkers.

Finally, as can be observed in Figure 8, almost 8 out of 10 young people absolutely distrust politicians. Specifically, 57.7% distrust social media, and 44.4% distrust media, only trusted by 6.1% and 7.6% respectively. Journalists did not obtain a better perception, with 4 out of 10 young people declaring distrust: they were only trusted by a percentage

of 19.2%. In addition, it can be observed that scientific and international institutions were trusted the most, with percentages of 64.7% and 45.3%, respectively.

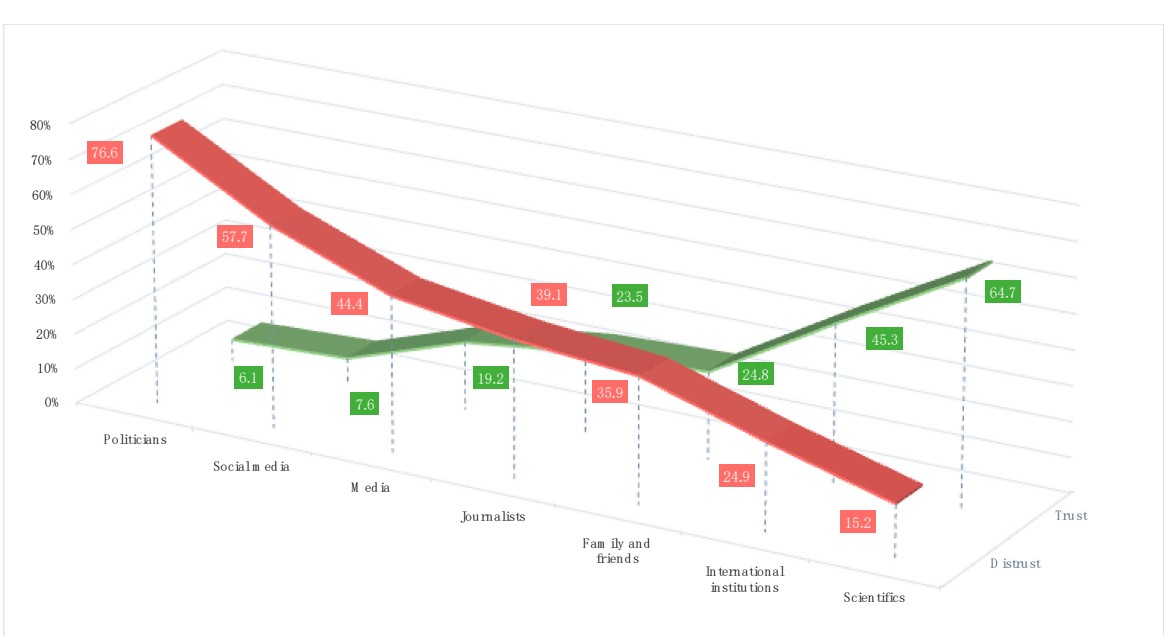

**Figure 8.** Percentages in trust and distrust feelings about social agents. Own elaboration.

## 5. Discussion and Conclusions

Misinformation and fake news have become a great global concern since the start of the COVID-19 pandemic, affecting all citizens, as recent literature indicates [5,8,10,17]. However, the manner in which it is reaching young people is extremely concerning, as shown in several studies [18,23,30,31]. It could be said that misinformation and fake news have been inherently human issues since the beginning of time [45], but the impact and the easy spread of the phenomenon through social networks call for urgent actions from universities and stakeholders (media and policymakers). Although the present research study does not offer a working hypothesis—as it is not intended to be an experimental research study, but a descriptive one—the main findings presented point to three different issues related to the results obtained, not only opening new lines of research but also providing benchmarks for specific actions from media stakeholders, policy makers, and educational institutions:

Firstly, regarding media consumption in line with other previous and recent studies such as Couldry, Livingstone, and Markham [40]; Jones [33]; and Mendiguren, Pérez, and Meso [18], 5 out of 10 young people declared never becoming informed by consuming the radio or the press, neither their analog nor online versions, while 7 out of 10 frequently or always became informed with social networks. These data coincide with previous studies and reports, but the interesting aspect offered by this study remains in the reliability rating given by the young people to the media they consume. Although theoretical arguments tell us that cognitive biases conduct us to believe what we see, before believing what we read, as assumed by Daniel Kahneman in his concept WYSIATI (What You See is All There is) [46,47], our sample is highly distrustful about the media that they consume the most, namely social networks and webpages, which tend to be more visual in nature. On the contrary, they truly believe in the reliability of media such as the radio or the press, which is highly interesting for media outlets, as it shows the need for new projects to focus on this unattended audience, as they once declared not consuming them.

Secondly, related to social media access and consumption, it is generally accepted in the literature that people would rather rely on the information that is immediately available, and the availability and the use of digital media are undeniable [5,40]. Moreover, the study

presented confirms how social media offering immediacy make them more appealing to Generation Z; however, a novel contribution to previous studies is the degree of awareness they showed regarding the lack of credibility of the content circulating through these media. The results on social network use—three-quarters of the sample used Instagram and WhatsApp the most—provided interesting data for media stakeholders. Perhaps in light of these descriptive data, media stakeholders should consider promoting specific content in these social networks to fight distrust and misinformation.

In third place, the data obtained, related to fake news, perceptions, and reception, allowed us to conclude that the Spanish Generation Z received and recognized fake news, but surely they did not use tools for their verification; i.e., 6 out of 10 young people in our study did not know what a fact-checker was. This lack of knowledge has been unfortunately confirmed by several studies [18,32] and reports [37,38]. Although they are conscious of the lack of credibility of social networks, they consume them intensively, assuming they constantly receive fake news that makes them feel manipulated, distrusted, and in danger. Several aspects or actions need to be brought together to provide a response to this situation, as highlighted by Lim and Tan [48], on the one hand, the Spanish government should foster significant, systematic, and comprehensive training programs in media literacy (following good practices from different countries such as Vietnam with their program "Fake≠Fact" or the UK's National Literacy Trust), and on the other hand, asking large corporations to collaborate in the fight against fake news. In this regard, we recently found that WhatsApp launched the "How WhatsApp can help you stay connected during the coronavirus (COVID-19) pandemic" program, which includes a step-by-step guide for users [49]. In light of the results obtained, we should add two more issues: first, media stakeholders are aware of young people and trust them, but they do not interact with them. They need to focus on young people as an audience, and they will pay attention if they feel that the media are also talking to them. Secondly, there is a need, more than ever, to reinforce media education at universities in order to promote and build up strong critical thinking skills in students. As shown by different data, the Spanish Generation Z does not lack media literacy skills in terms of access, use, and consumption, but the problem goes beyond traditional media literacy. As pointed out by Buckingham [50], rethinking media literacy should not be presented as an individual solution, as it relays responsibility to citizens and "absolves governments from their responsibility to solve problems" [50] (p. 230). Instead, re-thinking media literacy should be thought of as a global solution that involves governments, media stakeholders, and education leaders at schools and universities.

**Author Contributions:** Conceptualization, C.J.-N. and J.R.-R.; methodology, A.P.-E. and L.M.P.-E.; validation, A.P.-E. and L.M.P.-E.; formal analysis, A.P.-E. and L.M.P.-E.; investigation, A.P.-E., L.M.P.-E., C.J.-N. and J.R.-R.; writing—original draft preparation, A.P.-E.; writing—review and editing, A.P.-E. and L.M.P.-E.; visualization, A.P.-E.; supervision, L.M.P.-E. All authors have read and agreed to the published version of the manuscript.

**Funding:** This research received no external funding.

**Data Availability Statement:** The data presented in this study are available on request from the corresponding author. The data are not publicly available due to it is still an ongoing research.

**Conflicts of Interest:** The authors declare no conflict of interest.

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
