# Peer review of "Fake News Reaching Young People on Social Networks: Distrust Challenging Media Literacy"

_publications, doi:10.3390/publications9020024_

Round 1

Reviewer 1 Report

Overall, this manuscript is well structured and presents the results in a very clear and consistent manner.

However, there are opportunities for improvement in the following aspects:

.   The reference to Spanish media on page 3 line 110 is very ambiguous: does it include national, regional, municipal? Print, audiovisual and digital media?

. It should be pointed out, from the abstract, that the sample includes young Spaniards between 18 and 22 years of age who ARE STUDYING, since only undergraduate students are mentioned. There should also be some consideration of the percentages of this population that are not studying, and reasons should be given as to why this student population was chosen.

. The response rate of the instrument is not mentioned. 

In addition, some errors in punctuation (missing commas, missing quotation marks at the end of direct quotes, for example) and in sentences that seem to have been translated literally from Spanish and sound a bit forced in English need to be reviewed and corrected.

Author Response

Dear Reviewer, thank you very much for your comments, we have implemented all changes you suggested:

Minor changes:

Point 1: The reference to Spanish media on page 3 line 110 is very ambiguous: does it include national, regional, municipal? Print, audiovisual and digital media? 

Response 1: We absolutely agree with reviewer 1. We have added the report from where we retrieved data: Navigating the ‘Infodemic’: How People in Six Countries Access and Rate News and Information about Coronavirus. 2020. Oxford: Reuters Institute for the Study of Journalism

line 114: “According to Nielsen et al. report covering six European countries …”

Point 2: t should be pointed out, from the abstract, that the sample includes young Spaniards between 18 and 22 years of age who ARE STUDYING, since only undergraduate students are mentioned. There should also be some consideration of the percentages of this population that are not studying, and reasons should be given as to why this student population was chosen.

Response 2: It has been clarified in the Abstract that the sample was composed specifically with students

Line 15-16: “Focusing on a convenience sample of 408 Spanish young students from Z Generation aged from 18 to 22…”

Point 3: In addition, some errors in punctuation (missing commas, missing quotation marks at the end of direct quotes, for example) and in sentences that seem to have been translated literally from Spanish and sound a bit forced in English need to be reviewed and corrected.

Response 3: Several words and structures has been amended all along the manuscript following reviewer´s suggestion as can be seen in the “track-changes” version

Reviewer 2 Report

Very interesting and pertinent study, with the potential to be replicated in other countries. Just two brief notes for improvement:
Lines 35 and 86: change WHO reference, that is, explain what it means the first time this acronym is used;
Lines 99 and 100: it seems to us that the theoretical discussion, especially in this part, can be reinforced with some clues left precisely in a set of studies published in this journal in 2017: "News Consumption across Europe" (https://www.participations.org/ Volume% 2014 / Issue% 202 / contents.htm).

Author Response

Dear Reviewer, thank you very much indeed for your comments and your suggestions. 

Minor remarks: 

Point 1: Lines 35 and 86: change WHO reference, that is, explain what it means the first time this acronym is used;

Response 1: Absolutely agree, line 35 has been added: “called by World Health Organization (WHO)….”

Point 2: Lines 99 and 100: it seems to us that the theoretical discussion, especially in this part, can be reinforced with some clues left precisely in a set of studies published in this journal in 2017: "News Consumption across Europe" (https://www.participations.org/ Volume% 2014 / Issue% 202 / contents.htm).

Response 2: Thank you for introducing this work that has been added to our References and referenced in text, line 115

Adoni H, Peruško Z, Nossek H, Schrøder KC. Introduction: News consumption as a democratic resource – News media repertoires across Europe. Participations.Journal of audience and reception studies. 2017, 14(2):226-252.

English editing has been applied to all manuscript.

Kind regards